# LESS IS MORE: DISCOVERING CONCISE NETWORK EXPLANATIONS

**Neehar Kondapaneni**[1]**, Markus Marks**[1]**, Oisin Mac Aodha**[2]**, Pietro Perona**[1]
[1]Caltech    [2]University of Edinburgh

## ABSTRACT

We introduce Discovering Conceptual Network Explanations (DCNE), a new approach for generating human-comprehensible visual explanations to enhance the interpretability of deep neural image classifiers. Our method automatically finds visual explanations that are critical for discriminating between classes. This is achieved by simultaneously optimizing three criteria: the explanations should be few, diverse, and human-interpretable. Our approach builds on the recently introduced Concept Relevance Propagation (CRP) explainability method. While CRP is effective at describing individual neuronal activations, it generates too many concepts, which impacts human comprehension. Instead, DCNE selects the few most important explanations. We introduce a new evaluation dataset centered on the challenging task of classifying birds, enabling us to compare the alignment of DCNE's explanations to those of human expert-defined ones. Compared to existing eXplainable Artificial Intelligence (XAI) methods, DCNE has a desirable trade-off between conciseness and completeness when summarizing network explanations. It produces 1/30 of CRP's explanations while only resulting in a slight reduction in explanation quality. DCNE represents a step forward in making neural network decisions accessible and interpretable to humans, providing a valuable tool for both researchers and practitioners in XAI and model alignment. Project Page: `https://www.vision.caltech.edu/dcne/`

## 1 INTRODUCTION

Deep learning solutions are more likely to be trusted if their decisions are understandable to both expert and non-expert users Wang et al. (2020); Yu & Alì (2019). Motivated by this goal, the European Union has adopted regulations advocating an individual's right to an explanation in automated decision-making Goodman & Flaxman (2017). As deep learning systems find application in high-stakes domains such as medicine Bruckert et al. (2020); Tschandl et al. (2020); Rajpurkar et al. (2022) and autonomous driving Greenblatt (2016); Ni et al. (2020); Kun et al. (2018) there is increasing interest into the topic of eXplainable Artificial Intelligence (XAI).

A first approach to XAI is to develop decision algorithms that are more transparent by design Rudin (2019). A second approach is to compute post-hoc explanations on decisions

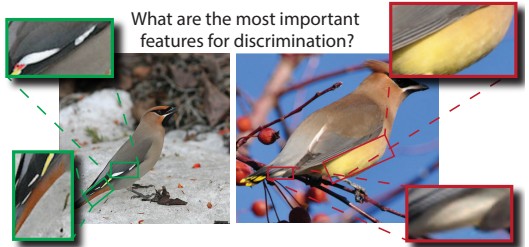

Figure 1: **Concise visual explanations.** What are the most important visual features to discriminate between Bohemian Waxwings (left) and Cedar Waxwings (right)? In this work, we explore if fine-grained features that align with those used by humans can be discovered from a trained neural network using XAI methods. Our method automatically extracts a concise set of explanations for individual images or at the class level.

made by already trained algorithms Xu et al. (2019). In this work, we pursue the second approach and focus our attention on image classifiers. Explaining predictions of existing models has the advantage of not imposing constraints on the types of models practitioners develop. Solutions to this problem can be broadly divided into *local* methods that explain a model's decision at the instance level (e.g., a specific patient's x-ray image) or *global* methods that explain a model's decisions at the

class level (e.g., a white spot in a lung x-ray is indicative of a disease) Das & Rad (2020). Recently, a family of *glocal* methods has been introduced to combine the best of both perspectives Schrouff et al. (2021); Achtibat et al. (2023).

A crucial concern for XAI methods is the ability of humans (experts or lay users) to digest the 'explanation' that a method provides. Humans have a limited 'perception budget' (famously posited as seven concepts in Miller (1956)). Thus, explanations must be simple enough to be understood and sufficiently few to comprehend. However, many methods generate a *single* attribution map that attempts to contain all the important visual features in one output Selvaraju et al. (2017); Lundberg & Lee (2017); Shrikumar et al. (2017). Such explanations tend to be too complex – humans are better at processing individual elements sequentially Pollock et al. (2002). In contrast, the recently proposed CRP Achtibat et al. (2023) method generates conditional attributions for individual neurons, yielding highly specific and spatially focused attribution maps that are relatively easy to understand. However, these maps are created for *each* neuron of the network. Thus, without any attempt to reduce the overall number of attribution maps, there are simply too many for a human to comprehend.

We propose a novel method called Discovering Conceptual Network Explanations (DCNE) that is both local and global and aims to produce concise explanations while retaining as much explanatory power as possible. We build upon CRP's Achtibat et al. (2023) conditional attribution maps by aggregating them across layers and reducing their dimensionality using Non-Negative Matrix Factorization (NNMF) Lee & Seung (1999). This enables us to reduce the number of explanations extracted from a model to 1/30 of its original size to a total of 10. This number is much closer to the 'perception budget' of humans.

To measure the quality of DCNE's explanations, we compared them to expert-defined feature masks that localize the important discriminative features of a species. We collect an annotated dataset with a subset of challenging bird species from the CUB dataset Wah et al. (2011) as a contribution of our study. We perform quantitative comparisons with a number of prominent existing XAI methods and find that our method's explanations are very concise while retaining most of the important information.

## 2 RELATED WORK

There are two main approaches to eXplainable AI (XAI) for deep neural networks, networks that are designed to be interpretable Chen et al. (2019); Koh et al. (2020); Poulin et al. (2006); Lin et al. (2014); Brendel & Bethge (2019); Bohle et al. (2021); Böhle et al. (2022); Donnelly et al. (2022); Koh et al. (2020) and methods that explain the network Bau et al. (2017); Fong et al. (2019); Morch et al. (1995); Sundararajan et al. (2017); Bach et al. (2015); Selvaraju et al. (2017); Lundberg & Lee (2017); Kim et al. (2018); Ghorbani et al. (2019); Ghorbani & Zou (2020); Goldstein et al. (2015); Akula et al. (2020); Chattopadhay et al. (2018); Srinivas & Fleuret (2019); Fu et al. (2020); Binder et al. (2016). Next, we discuss the most relevant existing XAI approaches through the lens of local, global, and glocal methods.

### 2.1 LOCAL METHODS

Local methods explain the important features in the input image that resulted in the network's prediction. Methods in this category use gradients (Selvaraju et al. (2017); Sundararajan et al. (2017); Simonyan & Zisserman (2015); Shrikumar et al. (2017)), modified gradients (Zeiler & Fergus (2014); Landecker et al. (2013); Bach et al. (2015); Montavon et al. (2017)), or perturbations (Lundberg & Lee (2017); Ribeiro et al. (2016; 2018)) to generate attribution maps on an *individual* input image (Guidotti et al. (2018)). Most of these methods generate only a single attribution map. It has been demonstrated that humans process complex information better when it is presented as individual elements Pollock et al. (2002). Single attribution maps are thus difficult to interpret because different, independent concepts are grouped together.

## 2.2 GLOBAL METHODS

Akula et al. (2020); Goldstein et al. (2015); Ghorbani et al. (2019); Kim et al. (2018) aim to discover concepts encoded within a model and explain how they relate to a class. These methods do not explain individual images but rather the features that describe a class broadly (e.g., stripiness is a feature that describes zebras). Kim et al. (2018); Schrouff et al. (2021); Goyal et al. (2019) require annotation in the form of user-defined sets of images that share concepts. Akula et al. (2020); Ghorbani et al. (2019); Goldstein et al. (2015) tried to remove this dependency by using clustering to group similar activation patterns. While global explanations provide a better conceptual understanding of model decisions, they do not generally provide explanations for specific images.

## 2.3 GLOBAL + LOCAL (GLOCAL) METHODS

Recently, Schrouff et al. (2021); Achtibat et al. (2023); Zhang et al. (2023); Fel et al. (2023) have attempted to combine both of the above families to understand network behavior better. Schrouff et al. (2021) introduced a method based on integrated gradients to measure a concept's importance in predicting a specific image instance. However, they do not have a method to visualize these concepts on the original image and require a curated set of images that define a concept to compute a concept activation vector. Achtibat et al. (2023) treat each neuron as an independent semantic feature detector and introduce the idea of conditional masks on the relevance flow to generate neuron-specific attribution maps on specific images. They also show how to use relevance scores to visualize the maximally relevant images for a specific neuron, thus showing the global concept a neuron encodes across several images. CRP has high explanation complexity, exceeding vastly the human "perception budget" Miller (1956). In this work, we improve upon CRP by using non-negative matrix factorization to discover a concise set of important features for each image and relate the features that make up a class across images.

## 3 METHOD

### 3.1 PROBLEM FORMULATION

Our goal is to develop a method that can explain all of the important semantic features in an image that a trained convolutional neural network uses to make its predictions at inference time. In addition, this method should be able to correlate these features across images while maintaining a low explanation complexity such that human users can understand the explanation with less effort.

We describe the methodological details in the following section while leaving all the implementation details to the appendix.

For a given image $x$, a trained neural network $F$ combines the outputs of many feature activations (i.e., neuron outputs) to produce an output, $F(x) = \hat{y}$. We denote each neuron in a network as $F_i^k$, where $k$ is the layer and $i$ is the neuron within the layer. We also denote $G$ as a local XAI method. For a given image and network, $G$ produces a local attribution for the image $a(x) = G(F(x))$. Some local XAI methods, e.g., Achtibat et al. (2023); Selvaraju et al. (2017); Ghorbani & Zou (2020), are able to condition their explanations on the layer and/or neuron. For these methods, we define a set of tuples $S_G = \{(k, i)\}$, which contains all the tuples of layers and neurons that $G$ is capable of producing attributions for. We generalize our notation for this setting such that

$$G(F_i^k(x)) = \{a(x, (k, i)) \in \mathbb{R}^{h \times w}\}, \tag{1}$$

where $i$ is ignored for settings where the method is only layer-specific. We define the final explanation for image $x$ as the set of all attributions

$$\mathcal{E}(x) = \{a(x, (k, i)) \, \forall \, (k, i) \in S_G\}. \tag{2}$$

The sets of image-specific explanations are aggregated into a final explanation, which is denoted as

$$\mathcal{E} = \{\mathcal{E}(x) \, \forall \, x\}. \tag{3}$$

We let $Q(\mathcal{E})$ represent the quality of an explanation $\mathcal{E}$ (see Sec. 4 for details). We also define the explanation complexity as the total number of attribution maps produced by a method, i.e., $|\mathcal{E}|$. Our goal is to produce a simpler, concise explanation $\mathcal{E}_s$, such that $Q(\mathcal{E}_s) \approx Q(\mathcal{E})$, while also having a reduced explanation complexity size $|\mathcal{E}_s| << |\mathcal{E}|$.

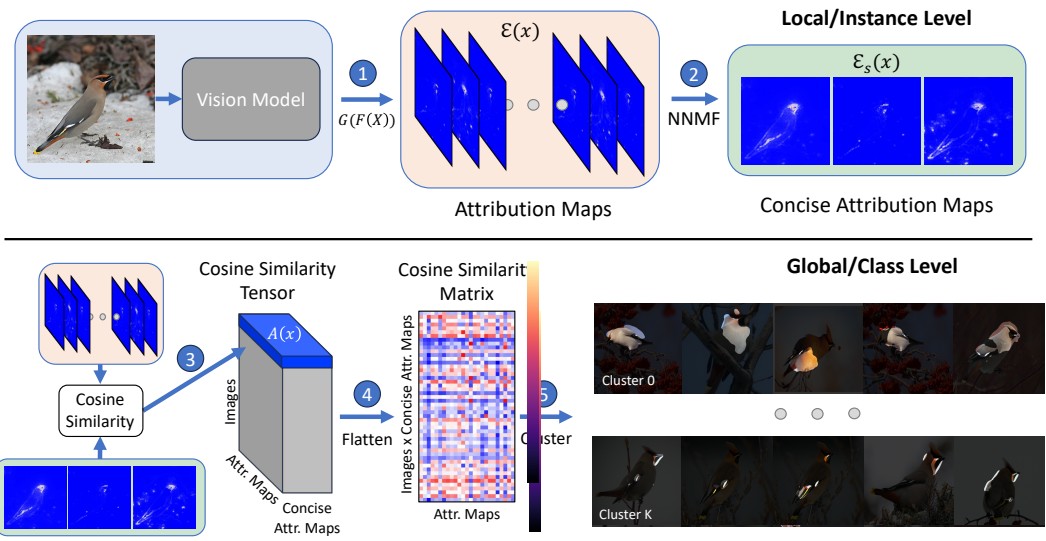

Figure 2: **Discovering concise attribution maps.** To create concise instance-level attribution maps, our method passes an image through a pre-trained CNN backbone (e.g., ResNet-34) and (1) generates the conditional attributions for the whole network through conditional masking (see appendix for details on CRP). (2) We reduce the many attribution maps produced by CRP to a smaller number using NNMF. (3) We then compute the cosine similarity of the concise attribution maps to the base attribution maps using cosine similarity. Since each base attribution map is generated from a neuron, the concise maps can be related back to a semantically tuned neuron. (4) We flatten the matrix along the image and attribution map dimension, and (5) perform clustering on the resulting matrix to generate global/class-level attribution maps, in which semantically similar image and attribution map pairs are grouped together.

## 3.2 CONCISE INSTANCE-LEVEL ATTRIBUTION MAPS

From here on, $G$ will refer to XAI methods capable of neuron (or layer) attributions. The set of conditional attribution maps $\mathcal{E}(\boldsymbol{x})$ produced by $G$ are able to indicate what image features different neurons (or layers) attend to in the original input image $\boldsymbol{x}$. Depending on network size, $\mathcal{E}(\boldsymbol{x})$ can be prohibitively large and prevent users from being able to search through them, i.e., the explanations can be of high complexity. This can hinder our ability to formulate a clear understanding of the important features being used by the network. For example, if $F$ is a ResNet-34 He et al. (2016), given the number of neurons in the network, for a single image, we would obtain $|\mathcal{E}(\boldsymbol{x})| \approx 8,000$.

We address the size issue by using non-negative matrix factorization (NNMF) Lee & Seung (2000) on the attribution maps in $\mathcal{E}(\boldsymbol{x})$ to produce $\mathcal{E}_s(\boldsymbol{x})$ (Fig. 2). Specifically, each attribution map $a(\boldsymbol{x}, (k, i)) \in \mathbb{R}^{h \times w}$ is flattened into a vector of length $d = h \times w$, producing a matrix $M_{\mathcal{E}(\boldsymbol{x})} \in \mathbb{R}^{|\mathcal{E}(\boldsymbol{x})| \times d}$. We use NNMF with $z$ components to decompose $M_{\mathcal{E}(\boldsymbol{x})}$ and keep the resulting $z \times d$ coefficient matrix. This coefficient matrix is reshaped back into images to produce each element of $\mathcal{E}_s(\boldsymbol{x})$, where $|\mathcal{E}_s(\boldsymbol{x})| = z$. We refer to the reshaped coefficient matrix as the concise attribution maps. In Fig. 6, we explore different numbers of concise attribution maps. We use $|\mathcal{E}_s(\boldsymbol{x})| = 10$ in all plots in the main paper. In practice, running NNMF on the complete set of attribution maps $\mathcal{E}(\boldsymbol{x})$ is costly, so we rank each attribution map by the sum attribution and select the top $n$ maps. In our experiments, we use CRP as the base method to generate conditional attribution maps. CRP is an explanation method that produces neuron-specific attributions by restricting relevance flow specifically through the neuron(s) of interest, generating disentangled neuron-specific attributions.

While on the surface related, our method is distinct from DFF Collins et al. (2018). In DFF, the authors pass $n$ images (from various classes) through a network and store the activations from the last convolutional layer of the network. This results in a tensor of size $n \times c \times h' \times w'$ which is reshaped into a $(n \times h' \times w') \times c$ matrix. Note that $h'$ and $w'$ here are not the original image size, but the output size of the last convolutional layer, which has $c$ channels. They perform NNMF on this

matrix to produce a reduced matrix with only $k$ components $(n \times h' \times w') \times k$. As later convolutional layers encode more semantic information, the pixels of equivalent semantic parts in different images share activity patterns in the same channels $c$. NNMF discovers this structure, producing coarse correspondence for shared parts between objects of different categories, for example heads, torsos, arms and legs across different people. In contrast, our method is focused on identifying diverse features from a large explanation set $\mathcal{E}(\boldsymbol{x})$ for a *single* image. We perform NNMF directly on the attribution maps of an explanation method rather than the network's activations, operating directly in the attribution space, not in the feature space.

### 3.3 Concise Class-Level Attribution Maps

The explanations outlined in the previous section are specific to *individual* image instances. However, we would also like to have a global explanation that summarizes the visual features that are common across all of the images in a class. Let $\mathcal{X}_c$ be the set of images all containing the same semantic class. Consider two images $(\boldsymbol{x}_1, \boldsymbol{x}_2) \in \mathcal{X}_c^2$, how can we relate the features discovered in $\boldsymbol{x}_1$ to features discovered in $\boldsymbol{x}_2$? Since the concise attribution maps are generated from a specific image $\boldsymbol{x}_1$, they retain the spatial structure of the object in that image and as the images $\boldsymbol{x}_1$ and $\boldsymbol{x}_2$ are different, we cannot directly compare the concise attribution maps with each other. In order to compare these attribution maps, the spatial information must be removed, and the attributions must be 'transferred' into a semantic space. For example, two attribution maps identifying a white wing patch on a bird should be clustered together irrespective of the spatial orientation of the birds. Neurons are semantically tuned, but the concise attribution maps are no longer directly related to any individual neuron. However, the base conditional attributions $\mathcal{E}(\boldsymbol{x})$ are neuron-specific and also image-specific. If we can relate the attributions from $\mathcal{E}_s(\boldsymbol{x})$ to $\mathcal{E}(\boldsymbol{x})$, we effectively project our image-specific concise attributions back into neuron space, making them comparable across images. A full discussion of the motivation of our clustering method is provided in Appendix B.

For an image $\boldsymbol{x}$, we compute the cosine similarity between explanation sets $\mathcal{E}_s(\boldsymbol{x})$ and $\mathcal{E}(\boldsymbol{x})$ producing a similarity matrix $A(\boldsymbol{x}) \in \mathbb{R}^{|\mathcal{E}_s(\boldsymbol{x})| \times |\mathcal{E}(\boldsymbol{x})|}$, where the entries are

$$A(\boldsymbol{x})_{ij} = \cos(\mathcal{E}_s^i(\boldsymbol{x}), \mathcal{E}^j(\boldsymbol{x}))$$

and where $i$ denotes the $i^{th}$ attribution map in $\mathcal{E}_s(\boldsymbol{x})$ and $j$ denotes the $j^{th}$ attribution map in $\mathcal{E}(\boldsymbol{x})$. We repeat this process for every image $\boldsymbol{x} \in \mathcal{X}_c$ to produce a similarity tensor $T \in \mathbb{R}^{|\mathcal{X}_c| \times |\mathcal{E}_s(\boldsymbol{x})| \times |\mathcal{E}(\boldsymbol{x})|}$. A visual depiction is available in Fig. 2. We flatten the first two dimensions of this tensor to produce a 2d matrix containing similarity scores between the concise attributions over all images and the neuron-specific conditional attribution in $\mathcal{E}(\boldsymbol{x})$. We then use a clustering objective on this similarity matrix to discover clusters (in practice, we run DBSCAN Ester et al. (1996)) over pairs of image and attribution maps. Concise attribution maps between and within an image can be clustered together according to how similar they are in neuron space, which is more semantically aligned than image space (Fig. 2). Importantly, this operation is distinct from maximum reference sampling in CRP. In that setting, maximum reference samples (typically eight) can be computed for *all* the neurons in a network, which results in a highly complex explanation. In contrast, our method produces a set of clusters for each class, each defining a discriminative feature used by the network. We define a cluster feature score to qualitatively assess a cluster's specificity to a particular feature. Since clusters are computed within a class, all the attribution maps for that cluster are compared to the set of expert-defined features for that class. For each concise attribution map in a cluster, we simply average the mean IoU computed for each expert-defined feature. This score reflects how specific a cluster is to one or more features. The number of clusters can be tuned by modifying the parameters of DBSCAN. In our experiments, DBSCAN produced at most seven clusters when using an epsilon value of 1.4 and a minimum cluster size of five. DBSCAN naturally detects and removes outliers, further reducing the noise in our clusters.

## 4 Evaluation

Evaluating explainable methods is challenging, as many evaluation criteria can be subjective and prone to confirmation bias Kim et al. (2022); Hoffman et al. (2018); Hesse et al. (2023). Moreover, commonly used evaluation criteria are defined via testing on naive, i.e., non-expert, subjects. Our application is a highly specialized task in which the interpretability of a network with expert-level

performance should be assessed by an expert, in our case, ornithologists, who understand the important features of the classes of interest. In order to address this problem, we use expert-defined features that allow us to repeatably evaluate many methods and images through the lens of a domain expert in a robust and relatively unbiased way. However, our evaluation does have limitations, which we discuss in 5.3. The next sections explain our evaluation methodology in detail.

## 4.1 FINE-GRAINED EXPLANATIONS DATASET

We evaluate our method on images from the CUB-200-2011 (CUB) dataset Wah et al. (2011). The original dataset consists of images from 200 different species of birds. While there are part-level and attribute-based annotations available in CUB that denote semantic parts and their state (e.g., the shape of a beak), these annotations alone are not sufficient to discriminate between very similar species pairs (e.g., like the birds in Fig. 1).

In order to generate a suitable evaluation set from CUB, we determine the discriminative semantic features for a subset of species using expert knowledge from the Cornell Lab of Ornithology's AllAboutBirds website. We annotated five classes from the CUB dataset: Bohemian Waxwings, Cedar Waxwings, White-Throated Sparrows, White-Crowned Sparrows, and Song Sparrows. For each species, we annotate 2-4 discriminative features based on the expert-generated identification guide from AllAboutBirds. For each image of a given class, we performed manual, pixel-level segmentation for each discriminative feature that is visible in that image, generating several feature masks per image. In total, we annotated 300 images, 60 images each for each of the five classes. We define $V_f(\boldsymbol{x})$ to be the set of expert-defined feature masks for an image $\boldsymbol{x}$ in class $\mathcal{X}_c$ where $f$ represents the index of the discriminative feature (Figs. 7,11).

## 4.2 MATCHING EXPERT FEATURES WITH ATTRIBUTION MAPS

To evaluate the quality of an explanation, $Q(\mathcal{E})$ for a specific image, at a specific threshold $t$, we compute the Intersection-over-Union (IoU) for each attribution map in $\mathcal{E}(\boldsymbol{x})$ against each feature mask in $V_f(\boldsymbol{x})$ and take the maximum score for each feature,

$$Q_f(\mathcal{E}(\boldsymbol{x}), t) = \max_{\{a(\boldsymbol{x}) \in \mathcal{E}(\boldsymbol{x})\}} \text{IoU}(a(\boldsymbol{x}) > t, V_f(\boldsymbol{x})) \tag{4}$$

$$Q_f(\mathcal{E}(\boldsymbol{x})) = \max_t Q_f(\mathcal{E}(\boldsymbol{x}), t) \tag{5}$$

We search for the best mean IoU for each attribution method on a held-out subset of the data over five threshold values of 0, 30, 50, 100, 150, 200, and 250 (masks are stored in uint8 format). The specific threshold used for each method and feature is presented in the appendix 1. We average over all images $\boldsymbol{x}$ to produce the final feature score for an explanation,

$$Q_f(\mathcal{E}) = \frac{1}{|\mathcal{X}_c|} \sum_{\boldsymbol{x} \in \mathcal{X}_c} Q_f(\mathcal{E}(\boldsymbol{x}))). \tag{6}$$

Under our evaluation protocol, increasing the number of attributions would most likely improve $Q_f(\mathcal{E})$, but would increase $|\mathcal{E}|$. Our method aims to find a balance between a diverse set of attributions and a low explanation complexity.

## 5 RESULTS

We evaluate DCNE on five bird species with semantic-level mask annotations based on expert knowledge described in Sec. 4. We compare the mean IoU scores for the features discovered by our method to the CRP Achtibat et al. (2023) baseline, Fullgrad Srinivas & Fleuret (2019), Gradcam Selvaraju et al. (2017), Layercam Jiang et al. (2021), and XGradCAM Fu et al. (2020). Since these methods do not perform neuron-specific attribution, to create a more comparable baseline, we generate attribution maps for each method conditioned on every convolutional layer in the network.

For Fullgrad, which naturally computes an attribution map for each layer, we simply include each of these attribution maps in the final set. For CRP, we include two variations on the explanation complexity: CRP-300 uses the attributions produced for each image by the 300 most relevant neurons

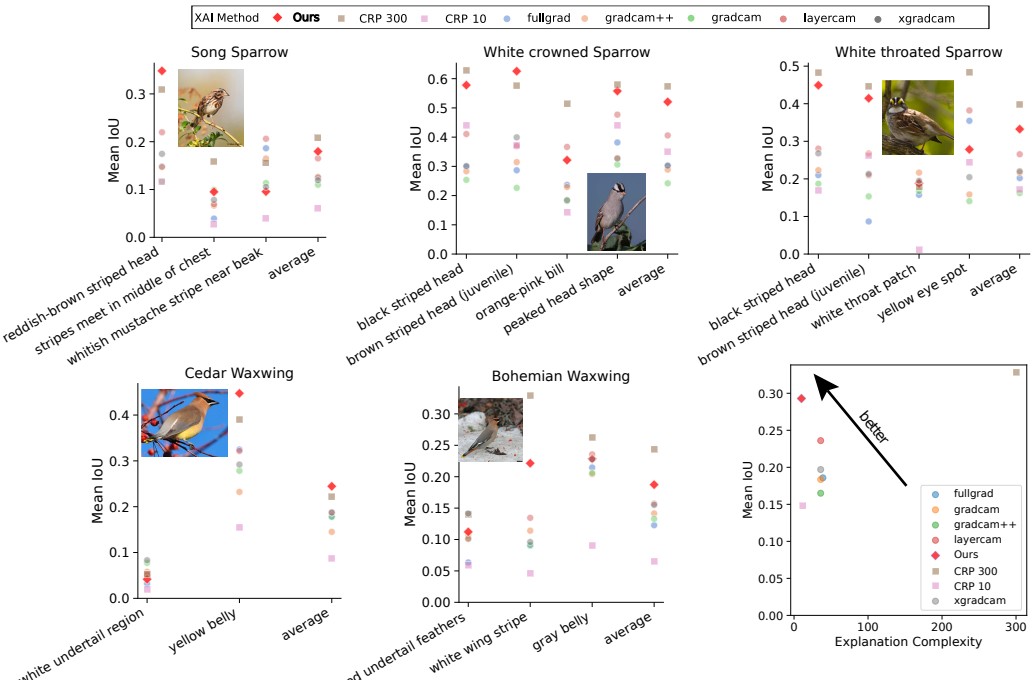

Figure 3: **Benchmarking instance-level attribution maps with expert-based features.** We introduce a new XAI benchmark for bird species expert feature matching. Each image is annotated with masks highlighting the birds' most important visual features, as defined by experts. Each subplot represents a different bird species. On the x-axis, we indicate the important features of that bird. On the y-axis, we show the mean IoU between that feature and our instance-level attribution maps (higher is better). The explanation complexity is indicated in the bottom right subplot. We compare our method against multiple baselines for five different bird species. In the bottom right plot, we see that our method has the second-highest mean IoU across all features and classes but maintains the lowest complexity.

(on average, over all the images of the class), and CRP-10 uses attributions from the top 10. Results are presented in Fig. 3.

On average, we find that CRP-300 has the highest mean IoU. This is not surprising since CRP-300 has access to the largest number of feature maps and is more likely to find a strong match to the expert-defined feature masks. However, there are several instances where CRP is worse than other methods. For example, in the whitish-mustache stripe of the song sparrow, the white throat patch of the white-throated sparrow, and the white-undertail region of the cedar waxwing, CRP-300 is outperformed by a CAM variant. The mean IoU for all these features tends to be on the lower side, implying that the network may not detect them at all or that they may always be encoded together with another feature, resulting in larger unions and lower mean IoUs. In particular, the white undertail of the Cedar Waxwing has low mean IoUs for all the methods we compare, implying the network is not detecting this feature at all. In addition, our method outperforms CRP-300 on the brown-striped head of the White Crowned Sparrow and the yellow belly of the Cedar Waxwing. It is possible that components produced by NNMF are entangling or disentangling features that can result in both increases or decreases in mean IoU (see Sec. 5.2 for details).

We also include results for CRP-10, which is constructed in the same way as CRP-300, but only uses the top 10 most relevant neurons and is directly comparable to the 10 components produced by NNMF. We observe that CRP-10 is significantly worse in all classes and features. Qualitatively, one can see that the top most relevant neurons tend to produce redundant attributions that fixate on the same feature in the image (Fig. 12). We find that our method has the second-highest mean IoU (averaged across all classes) while requiring only 1/30 of its explanation complexity.

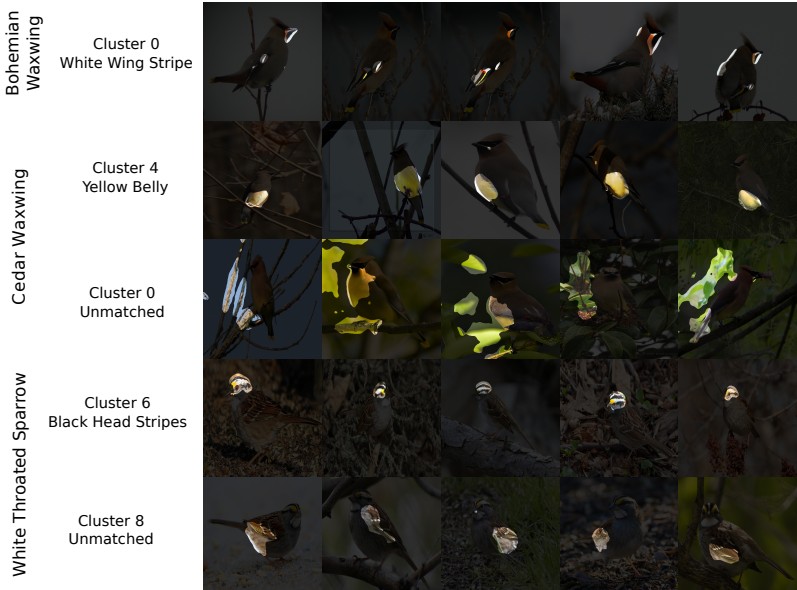

Figure 4: **Selected clusters**. We select several of the clusters (rows) generated by our method to demonstrate interesting properties of the global concepts discovered. For each cluster, we select five image and attribution map pairs that are closest to the centroid of the cluster (left-to-right). Cluster 0 of the Bohemian Waxwing (BW), Cluster 4 of the Cedar Waxwing, and Cluster 6 of the White Throated Sparrow (WTS), align well with several expert-defined features. However, our method discovers other semantically consistent features that do not match with any of the selected expert-defined features. We use expert descriptions from AllAboutBirds to evaluate whether these features are sensible. For Cluster 0 of CW, we find a description that indicates "foraging birds often perch acrobatically at the tips of thin branches to reach fruit" and for Cluster 8 of WTS we find "two white wingbars on rich reddish brown wings".

## 5.1 CLASS-LEVEL FEATURES VIA CLUSTERING

We take all of the components produced by NNMF (for all images in a class), compute the components' cosine similarity to the neuron attribution maps, and use DBSCAN to cluster the components based on this similarity matrix (see Sec. 3.3 for details). For each class, we find that 1-2 clusters are well-aligned with our expert-defined features. We show all of the cluster similarity matrices in the appendix.

After computing the feature clusters, we measure the average IoU score of each image in a cluster for each feature (see Fig. 5A and B for examples). The cluster feature scores (see details in Sec. 3.3) indicate which expert-defined features best align with the cluster. Qualitatively, we find that some discovered clusters align well with the expert-defined features that are present in the dataset (see Fig. 4). Cluster 0 of the Bohemian Waxwing aligns with the "White wing stripes" expert-feature for that class. Cluster 4 of the Cedar Waxwing aligns with the "Yellow belly" expert-feature for that class. Cluster 6 of the White Throated Sparrow aligns with the "Black head stripes" expert-feature for that class. We notice that the features in each cluster are very diverse. The "White wing stripes" feature is small in scale and can be spread across multiple non-linked masks, while the "Yellow belly" feature is comparatively large in scale. Since we have more clusters per class than expert-based features, we end up with clusters that do not match any of the expert-based features. After inspecting these closely, we find, to our surprise, that some of them encode semantically meaningful features that are described on AllAboutBirds but we did not annotate for.

## 5.2 (DIS-)ENTANGLED ATTRIBUTION MAPS

Since the network is trained only on the class labels, there is no guarantee that the features it uses for classification align with the expert-based masks. Fig. 5A illustrates this case with the Bohemian Waxwing feature "White wing stripes". This expert-based feature consists of two masks for the

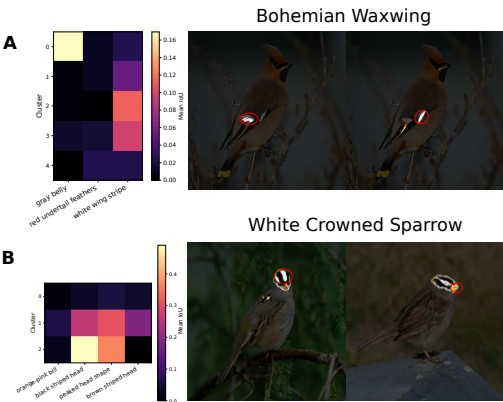

Figure 5: **Entangled and disentangled features**. (**Left**) We show the confusion matrix between the discovered clusters and the features we annotate. (**Right**) we show attribution maps generated by our method. In panel **A**, we show features that are grouped together in the 'expert' feature masks but are disentangled in our explanations. We can see that the explanation method separates the white wing stripes into two separate concepts. In panel **B**, we show features that are consistently entangled. The orange beak and the black-striped crown are treated as separate features in the evaluation based on the expert's description, but the explanation suggests they are encoded together.

two white stripes on the wing of the bird combined together. While this expert-based feature is generally discovered well, it is disentangled into two attributions, one per white stripe. In this case, our intersection would be lower, decreasing the IoU, even though the network is discovering all of the important pixels. Fig. 5B shows the reverse, where Cluster 1 entangles two features of the White Crowned Sparrow, the orange beak and the black-striped crown. In this case, our union with the expert-based masks would be higher, decreasing the resulting IoU.

### 5.3 LIMITATIONS

Our evaluation methodology has some limitations. For methods to be interpretable to humans, the explanation of the model's predictions should align with the visual features humans use. However, there is no guarantee that networks use human-aligned concepts to process information. Additionally, the concise attribution maps may dis/entangle expert-based features, causing a decrease in IoU (see Sec. 5). This limitation may be partially overcome by either advancing the underlying models by explicitly decomposing the attributions into disentangled components.

To evaluate our method, we introduced a new dataset and task, which we deem appropriate for the problem we are describing. Nonetheless, we would like to include classical interpretability benchmarks in future studies as well, such as justified trust and explanation satisfaction Hoffman et al. (2018), pointing games Zhang et al. (2018), or HIVE Kim et al. (2022). Moreover, future work could include additional datasets and other neural network architectures, e.g., visual transformers Dosovitskiy et al. (2021).

## 6 CONCLUSION

We presented DCNE, a new approach for generating a concise set of semantic visual features from a trained deep image classifier. DCNE reduces explanation complexity while retaining explanation quality. To quantify this, we created a new evaluation dataset by annotating closely related bird species with expert-defined discriminative feature masks. We measured the alignment between the features produced by different XAI methods and the human expert-defined feature masks. We found that DCNE achieved the best trade-off between performance and complexity, with the second-highest performance after the CRP-300 method, at only 1/30 of the complexity. We then described a clustering-based method that creates class-level features and demonstrated that many of the clusters match the expert-based annotations. Surprisingly, we also found unannotated discriminative features as well, all while maintaining a low explanation complexity.

In Mac Aodha et al. (2018), the authors showed that visual explanations are useful aids for teaching concepts to human learners, but only used simple methods to generate single local explanations. In the future, we would like to evaluate whether our method, which balances explanation complexity and quality, can improve on their results and potentially be used to teach more complex visual categorization tasks. Finally, our evaluation dataset could be used to measure the alignment between model-learned concepts and human expert concepts.

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

# A    APPENDIX

## A.1    ABLATION EXPERIMENTS

We perform ablation experiments investigating the impact of specific design choices.

**NNMF Components.** In Fig. 6 A we sweep over the number of components used by NNMF. Starting from the CRP-300 base set, we test 3 to 250 components. We run NNMF for a maximum of 200 iterations, with early stopping if it converges. We observe that the number of components does matter. For most classes,

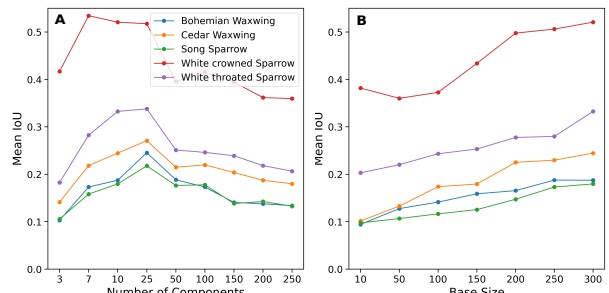

Figure 6: **Paramater Sweeps**. We explore a range of parameters for the number of NNMF components and the CRP base size.

increasing the number of components up to 25 improves the fit. Beyond that, the alignment between the components and human-defined features decreases. We use 10 components in our work since it is near the perceptual budget Miller (1956).

**Base Size.** In Fig. 6 B we sweep over the base set size. We evaluate 10 (comparable to our method), 50 (comparable to CAM per convolutional layer methods), 100, 150, 200, 250 and 300 attribution maps in an explanation set $\mathcal{E}(\boldsymbol{x})$ for an image. We run NNMF with 10 components on all of these base sets. We find that increasing the base set size improves mean IoU performance for all classes.

## A.2    TERMINOLOGY

In the main text, we use several different terms. In this section, we elaborate on their meaning and give visual examples.

**Expert-defined feature masks.** We use the term expert-defined feature masks to describe the ground truth segmentation masks produced by the annotator (Fig. 7). These binary masks are based on features defined by experts as important diagnostic attributes for each bird species. We use the word 'masks' as these annotations are binary (attribution maps are not).

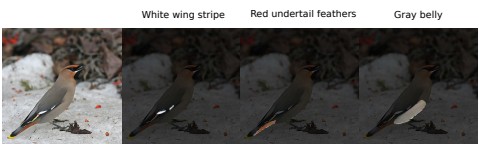

Figure 7: **Expert-defined feature masks.** Visualizations of the annotators' ground truth masks for the Cornell Lab of Ornithology specified features.

**Activations.** Activations are computed on the forward pass and are the outputs of a given layer of the network (Fig. 8). In 2D convolutional networks, they are known as both activations and feature maps. For example, in a ResNet-34, the last layer produces an activation map (or feature map) of size Nx512x14x14.

**Attributions.** Attribution maps indicate what the network used in making its decisions and are computed in the backward pass (Fig. 9). In all the algorithms compared in this work, the attribution maps are generated on the input space and are equal in width and height to the input image.

**Conditional Attributions.** Concept Relevance Propagation introduces the idea of conditional attributions and is a generalization of Layer-wise Relevance Propagation (Fig. 10). Conditional attributions visualize the attribution maps conditioned on a set of neurons the user is interested in. To do this, they modify the relevance flow with a conditional mask such that it is only passed through the neurons of interest in a given layer. Further details can be found in Achtibat et al. (2023).

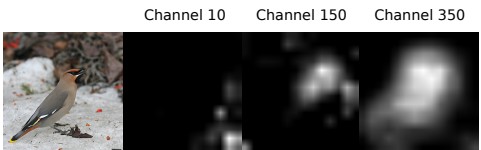

Figure 8: **Activation maps for ResNet-34.** Visualizations of the outputs of the last convolutional layer. There are 512 channel activation maps of 14x14 pixels. We show three activation maps, upsampled to the original input image resolution (448, 448): channel 10, channel 150, and channel 350. Each channel is activated differently, capturing different features of the original image.

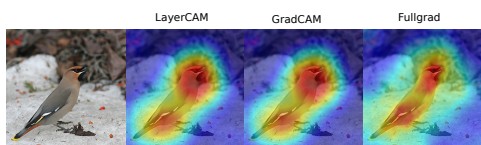

Figure 9: **CAM variant attributions.** CAM attributions are generated by upsampling the importance-weighted activations from the last layer of a network. Therefore, the CAM attributions tend to be significantly coarser than CRP's. The importance weights are usually computed using some form of modified gradients.

## B  INTUITION FOR OUR CONCISE CLASS-LEVEL ATTRIBUTION MAPS

The goal of our clustering operation is to discover a small set of paired images and attributions that can summarize all of the *class*-specific features used by the model. In this section we provide the intuition behind our clustering approach.

Our base attribution method (CRP-300) selects the 300 most relevant neurons across all of the images in a class. We use CRP to generate conditional attributions for each of these neurons for each image. Starting from this base method, this leaves us with two matrices that could be considered for clustering. First, an $N \times 300$ matrix, $M_1$, where each row is an image and the columns are the neurons' relevances. Second, an $N \times 300 \times (h \times w)$ tensor of images and attributions, where we would flatten the last two dimensions of the tensor to create an $N \times (300 \times h \times w)$ matrix, $M_2$.

If we cluster the $M_1$ matrix, we would group semantically related images, but would not be able to group based on visual features. If we instead cluster the $M_2$ matrix, we would group visually similar attribution maps but lose semantic information, i.e., the same feature, oriented differently, could not be grouped together. What we really want is a method that groups semantically similar pairs of images and attribution maps.

To better understand our approach, it is easiest to conceptually pair each image and attribution map and treat them as independent entities. As a simple example to ground intuition, lets consider two images with two attribution maps each: $a(\mathbf{x_1}, 1)$, $a(\mathbf{x_1}, 2)$, $a(\mathbf{x_2}, 1)$, $a(\mathbf{x_2}, 2)$. Suppose that $\mathbf{x_1}$ and $\mathbf{x_2}$ contain the species 'Bohemian Waxwings' but the birds are oriented differently in each of the images, e.g., in $\mathbf{x_1}$ it faces the right and in $\mathbf{x_2}$ it faces the left. In addition, the features represented by $a(\mathbf{x_1}, 1), a(\mathbf{x_1}, 2), a(\mathbf{x_2}, 1), a(\mathbf{x_2}, 2)$ are all the white wing stripe (recall that there may be redundancy in the attribution maps for a specific image). How can we group semantically related attribution maps within a single image, but also across two different images?

Within a single image, we compute cosine similarities from each image and attribution pair to every other image and attribution pair generating a cosine similarity matrix for that image. Thus, visually similar pairs, e.g., $a(\mathbf{x_1}, 1)$ and $a(\mathbf{x_1}, 2)$, would have a high cosine similarity. Importantly, since our attributions have a one-to-one mapping to neurons, it enables us to project attributions back into the *semantic* neuron space. For example, an attribution map for the white wing stripe will have high cosine similarity to all the neurons that attribute their activations to the white wing stripe.

When comparing across images, neurons that react to the white wing stripe of the Bohemian Waxwing should do so for both $\mathbf{x_1}$ and $\mathbf{x_2}$ generating similar patterns of within image cosine sim-

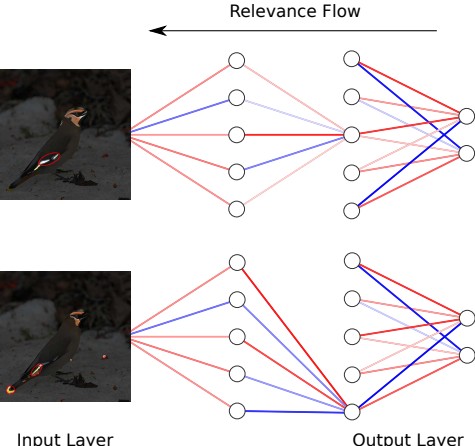

Figure 10: **Conditional Masking Generates Different Attributions.** We show a toy schematic to demonstrate the concept of conditional masking Achtibat et al. (2023). The relevance flow is restricted through a specific neuron in the second hidden layer, resulting in unique attributions for that neuron. For example, the first neuron may focus more on the white stripes, whereas the second focuses on the yellow tags on the wings.

ilarities (see Fig. 2 Cosine Similarity Matrix). The clustering algorithm finds these patterns and is able to group image and attribution pairs according to their semantic information. In practice, we use this method with our (10) *concise* attribution maps. We first generate a tensor of cosine similarities of $N \times 10 \times 300$, which would be flattened into a matrix of $(N \times 10) \times 300$. Each row of this flattened tensor is an image and attribution pair and the columns are the neurons' similarity score to that pair. Then we cluster this matrix to find semantically similar image and attribution pairs.

## C  IMPLEMENTATION DETAILS

### C.1  MODEL TRAINING

We train a ResNet-34 on the CUB dataset. We randomly split the data into training (70%), validation (15%), and test sets (15%). We train for 95 epochs using stochastic gradient descent with learning rate 1e-4, weight decay 1e-4, and momentum 0.9. We select the model with the lowest validation loss. The final model had a test accuracy of 0.818% over all 200 classes. We choose several classes that humans commonly misidentify, according to the iNaturalist website iNaturalist. The classes and their class-specific test accuracies are Bohemian Waxwing (89.4%), Cedar Waxwing (100%), Song Sparrow (77.3%), White-crowned Sparrow (100%), and White-throated Sparrow (93.3%).

### C.2  ATTRIBUTION METHODS

To generate the CAM-based visualizations, we use Gildenblat & contributors with a slight modification to the code for FullGrad Srinivas & Fleuret (2019) to save every layer's output. To generate the CRP attribution maps, we use the repository published by the authors Achtibat et al. (2023). We use the sklearn Pedregosa et al. (2011) implementation for non-negative matrix factorization with 3, 5, 10, or 20 components to generate the concise attribution maps. For 10 components and 300 base attribution maps, NMF roughly takes 45 seconds on an AMD Ryzen 7 3700X 8-Core Processor. We use an NVIDIA GeForce TitanX for training the model and generating attributions. We search a list of thresholds 0, 25, 50, 100, 150, 200, and 250 for each method and each species that maximizes that method's mean IoU on a held-out subset. We indicate these threshold values in Table 1.

### C.3  GROUND TRUTH ANNOTATIONS

We used Upwork to hire an experienced annotator at a flat fee of $125 USD to annotate 300 images (60 images per class). Annotations were performed using the open-source software label-

| | Bohemian Waxwing | Cedar Waxwing | Song Sparrow | White Crowned Sparrow | White Throated Sparrow |
|---|---|---|---|---|---|
| CRP-300 | 50 | 25 | 100 | 50 | 50 |
| CRP-10 | 25 | 0 | 100 | 50 | 100 |
| Ours | 50 | 25 | 50 | 25 | 50 |
| Fullgrad | 150 | 150 | 200 | 200 | 200 |
| GradCAM | 150 | 100 | 200 | 200 | 200 |
| LayerCAM | 100 | 50 | 150 | 100 | 100 |

Table 1: **Thresholds for method and species.** We choose a threshold value from a list (0, 25, 50, 100, 150, 200, 250) that maximizes the method's meanIoU on a held-out subset of a particular class.

studio Tkachenko et al. (2020), specifically the brush segmentation tool. The annotator was given clear instructions, developed from the allaboutbirds AllAboutBirds guide, to annotate 2-4 parts (depending on the species). The annotator was instructed not to guess its location if the part was not clearly visible in the image. We reviewed each image and asked for corrections to images where the annotator made mistakes. After we were satisfied with the results, the work was approved. No algorithms were tested between the annotation, review, and approval processes.

# D  ADDITIONAL VISUALIZATIONS

## D.1  COMPARING CRP-10 TO DCNE[10]-300

We show visualizations of the top-10 CRP attribution maps compared to the top-10 DCNE attribution maps (Fig. 12). CRP-10 visualizations are more redundant and do not capture all of the different features used by the network. We also see a possible limitation of this method: the concise attributions will sometimes attribute very strongly to the background making them more difficult to interpret.

## D.2  CLUSTER VISUALIZATIONS

One of the benefits of this method is that the explanations are concise. We show that the entire class can be summarized into 4 - 10 clusters that convey the properties used by the network in predicting the class of interest (Figs. 13,14,1516,17). In general, 1-2 clusters align well with the features we annotated. Some remaining clusters have clear semantic meanings but may not have been annotated. Other clusters are less interpretable.

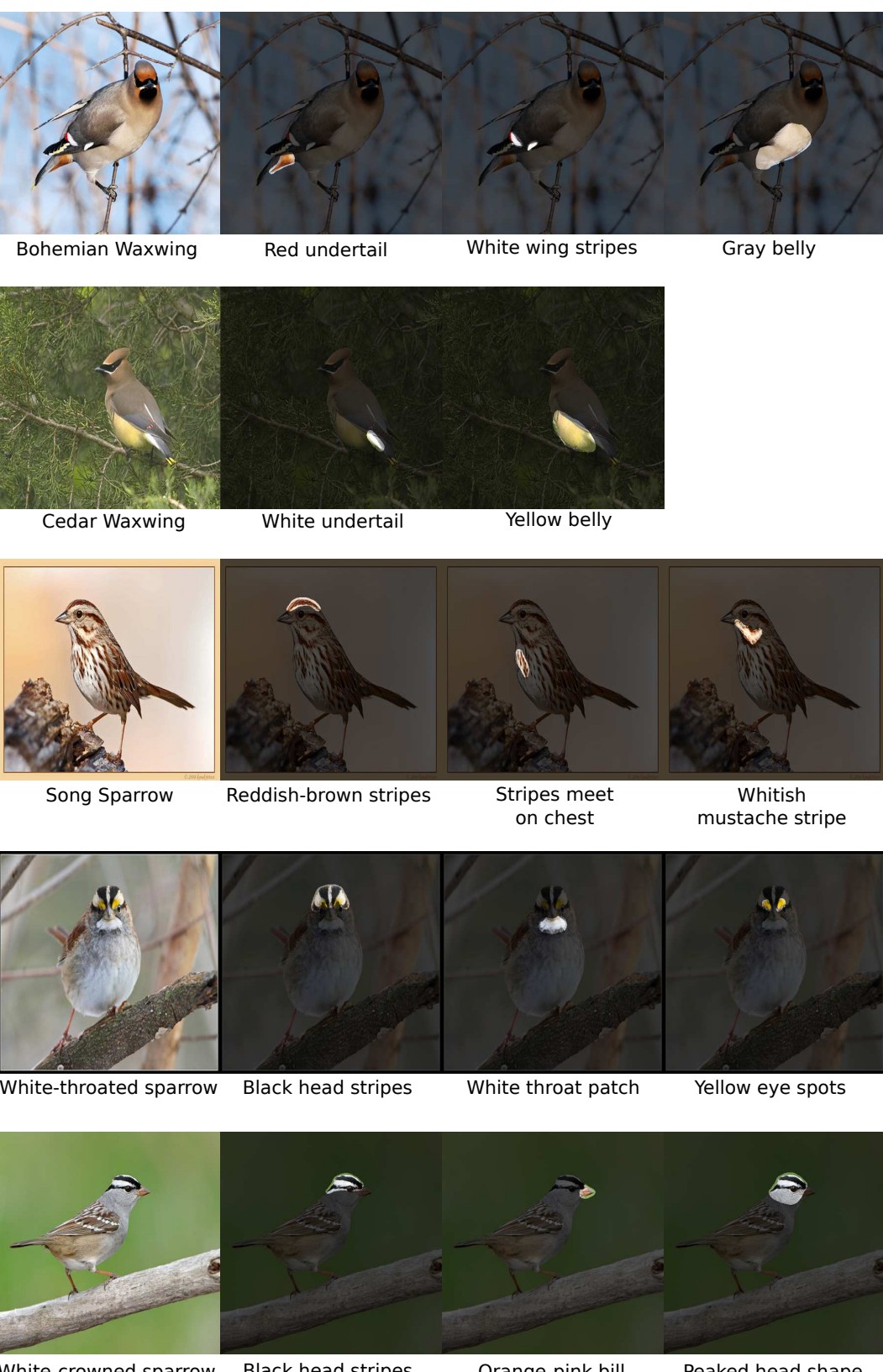

Figure 11: **Sample expert annotations.** We show one bird per class and the features that were annotated for that class.

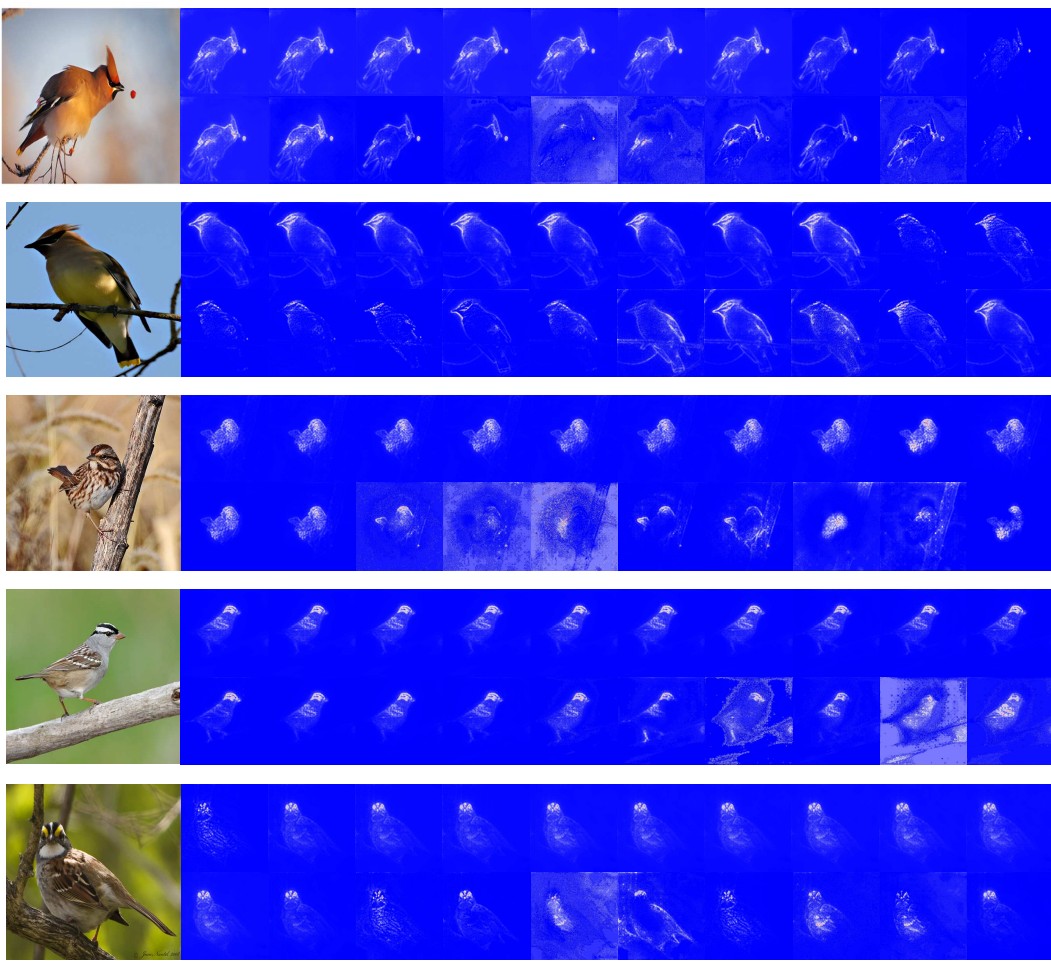

Figure 12: **Comparing CRP-10 and DCNE-10.** In each row, we have an image of a bird from each class. In the top panel, we show the attribution maps from the top 10 most relevant neurons (on average over the whole class). We show the 10 concise attribution maps generated from the CRP-300 set on the bottom panel. The concise attribution maps highlight different features. In contrast, the top 10 CRP attribution maps are more similar to each other and activate more generally across the whole bird.

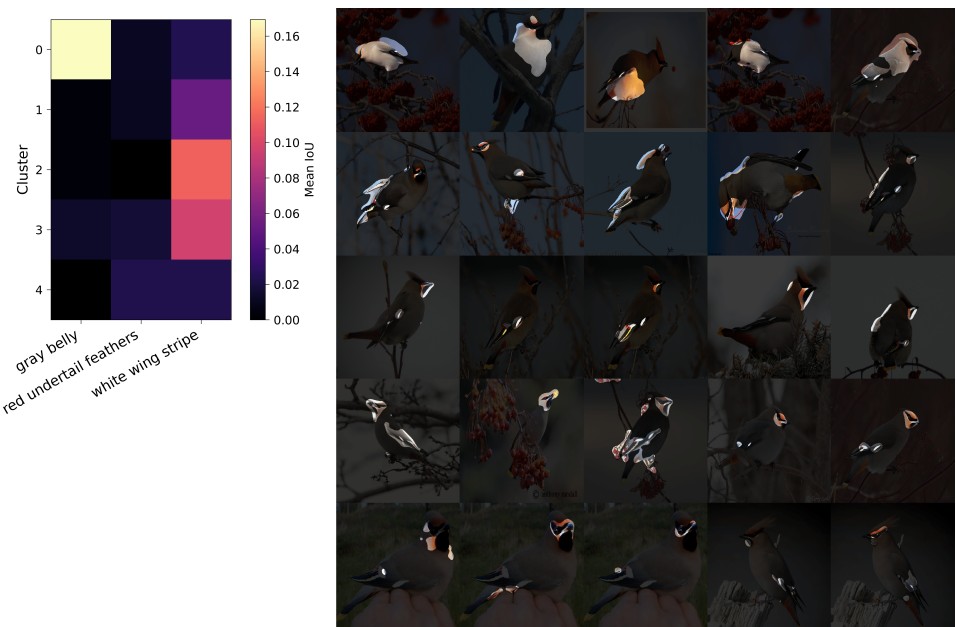

Figure 13: **All the clusters of the Bohemian Waxwing.**

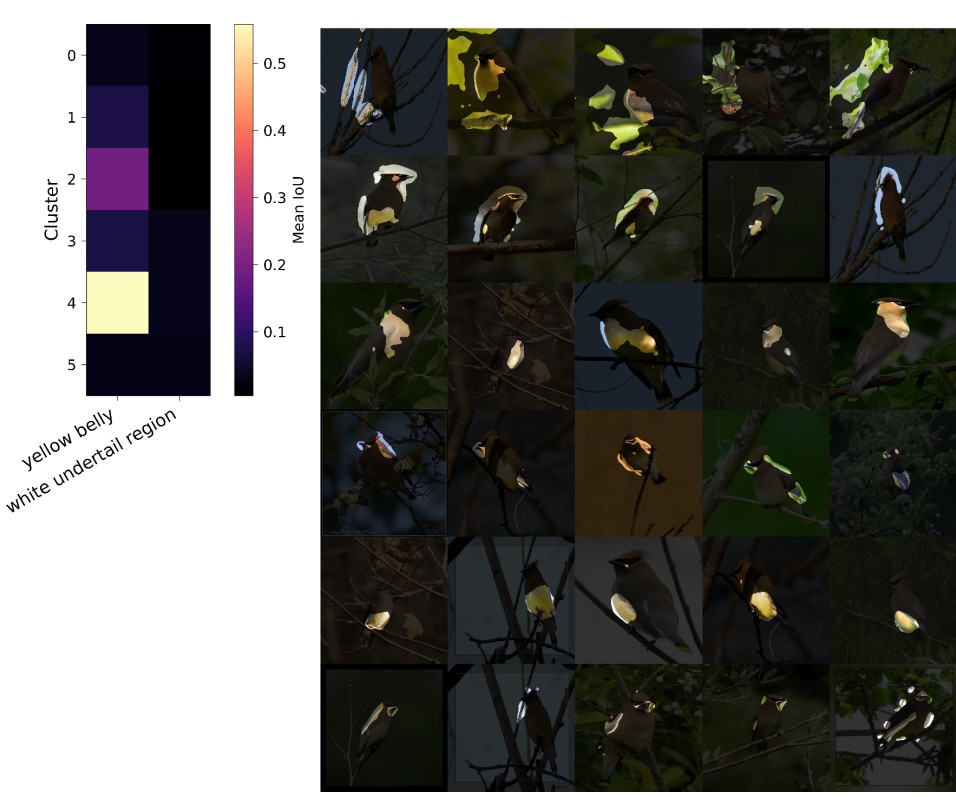

Figure 14: **All the clusters of the Cedar Waxwing.**

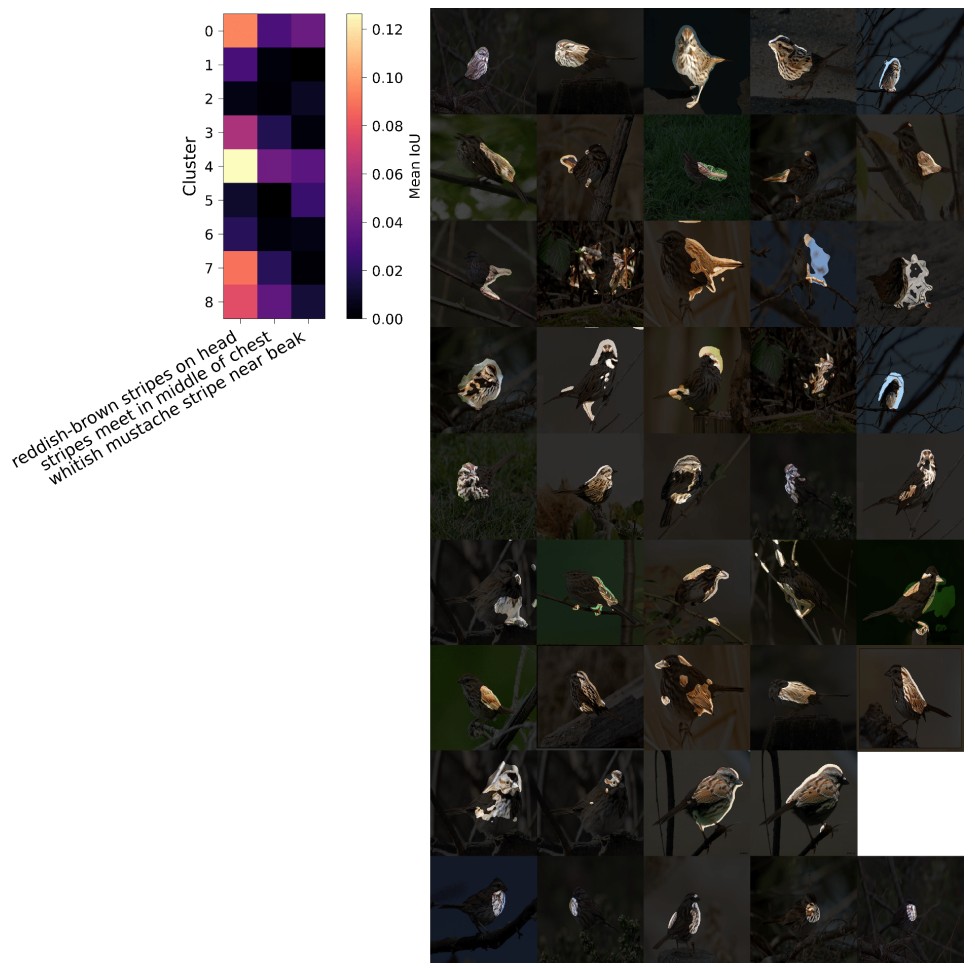

Figure 15: **All the clusters of the Song Sparrow.**

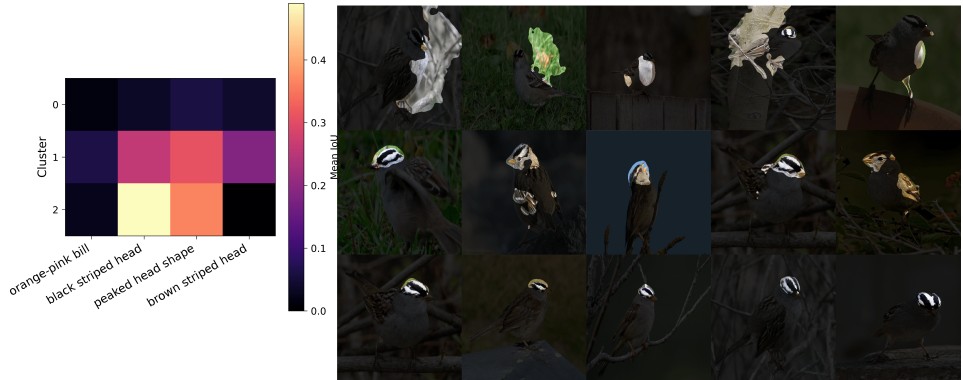

Figure 16: **All the clusters of the White-Crowned Sparrow.**

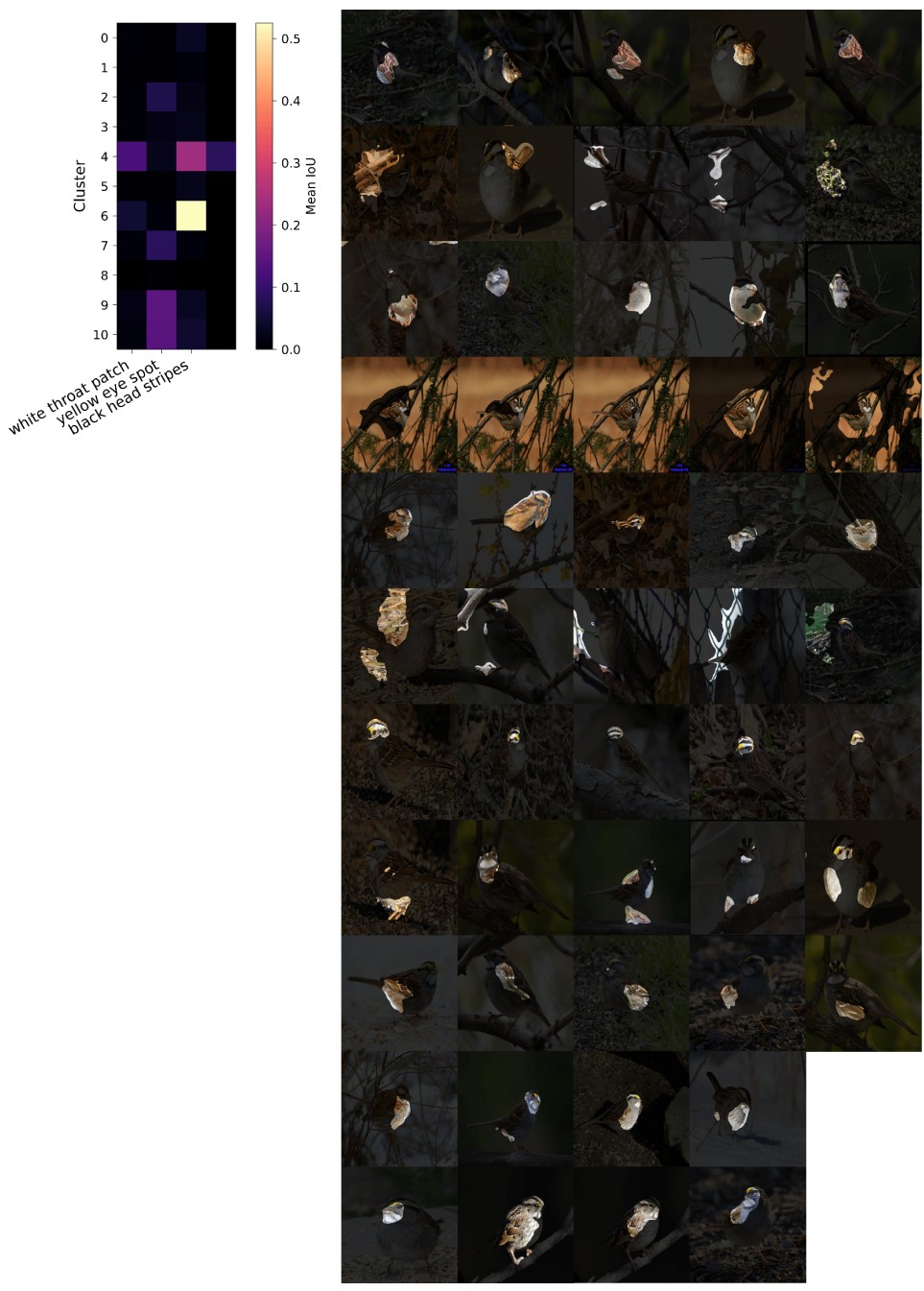

Figure 17: **All the clusters of the White-Throated Sparrow.**

