# OpenReview forum: "Less is More: Discovering Concise Network Explanations"
_ICLR.cc/2024/Workshop/Re-Align — ICLR 2024 Workshop Re-Align Poster_

### Official Review · Reviewer_tC4L · 2024-02-20
**The paper introduces DCNE, a concise, low-complexity method for explaining ConvNN predictions. The method 1) provides only limited insight into the operations of the NN and 2) needs more comparison on other benchmarks.**

**Rating:** 2
**Fit:** 3
**Confidence:** 2

**Workshop Review:**

This paper introduces DCNE, a method that aims to explain all of the semantic features from an image that a convNN uses to make a prediction. This method aims to provide concise and low-complexity explanations in the form of feature maps.
The paper starts by defining local, global, and “glocal” methods.
These definitions are followed by the problem formulation of computing explanations that are both descriptive of a neural network's prediction at inference time but still low on complexity so as not to exhaust the “perception budget” of the human.
The author describes the DCNE more thoroughly and places it in the larger landscape of attribution methods.
DCNE is evaluated on the CUB-200-2011 dataset. The method produces high-performing explanations that are prone to be entangled at a fraction of the complexity compared to other SOTA methods.

Pros:
- The paper is well written, and it is easy to follow the author's design decisions to develop DCNE.
- The authors address the problem of entanglement with its methods (which is one of the biggest problems in interpretability) – I greatly appreciate this transparency.

Cons:
- They mention it in the limitation section, but it feels understated that attribution methods only vaguely hint at what computations actually happen in the neural network. Since the method doesn’t do causal interventions on the network, there is no certainty that the explanations faithfully describe what is happening in the forward pass of the convolutional network. This is worth pointing out because interpretability aims to reassure that a network is doing “the correct thing,” which is not given in the attribution method.
- Lack of comparison to other benchmarks and datasets

Overall, I find the ideas in this paper more compelling than its limitations.

**Reason For Not Giving Higher Score:**

- more comparisons
- better solutions for shortcomings of the method

**Reason For Not Giving Lower Score:**

- well written
- addresses important questions for interpretability

**Reviewer Domain:**

machine learning

---

### Official Review · Reviewer_Ji7k · 2024-02-23
**Borderline**

**Rating:** 2
**Fit:** 2
**Confidence:** 2

**Workshop Review:**

Firstly, congratulations to the authors on their work. The paper addresses a compelling topic, and I am eager to see a revised full version presented at a conference. The results depicted in Figure 4 are particularly convincing and strengthen the case for accepting the paper. However, I must note that it's on the borderline, and I have several points to discuss, categorized as major (M) and minor (m) concerns.

**M.1**: The method appears quite complex with numerous hyperparameters involved. It entails computing neuron importance scores, applying attribution methods to the top-k neuron importance scores, performing NMF on the attributions, and finally utilizing DBSCAN on the NMF embedding. The presence of multiple hyperparameters, including those for selecting the top-k neurons, defining importance in the neuron space, NMF, and DBSCAN, adds to the method's complexity.

**M.2**: The evaluation of the method is not convincing. While the authors acknowledge the limitations of their evaluation, it does not exempt them to provide good evaluation for their method.
Two primary issues arise: (1), the metric $Q_f$ compares attributions that best match human annotations for their hundreds of attributions against single attribution maps produced by other methods, which may lead to unfair comparisons. Secondly, as noted by Adebayo et al.[1], "It's not because an explanation make sense that it reflect the evidence for the prediction", the evaluation primarily measures model alignment rather than the effectiveness of the method itself. This discrepancy is often called plausibility vs fidelity, see Jacovi & al [2] or Colin & al [3].

To address these concerns, I propose moving the current evaluation to supplementary material and conducting a human evaluation similar to Colin et al.[3] or HIVE[4] -- as you proposed at some point! Additionally, utilizing metrics such as deletion[5], insertion[5] or $\mu$Fidelity[6], as implemented by Quantus[7], could provide further insights.

Now, onto minor issues:

**m.1**: The assumption of positivity in attributions with NMF raises questions about handling non-positive attributions.

**m.2**: Also regarding **M.1**, providing a pseudo-algorithm for the method would enhance clarity.

**m.3**: You are likely aware of the superposition problem: neuron-wise explanation does not inherently make more sense than any other vector $v$ in activation space. A discussion of this limitation in the paper's section on limitations would be beneficial. However, given that you study all the neurons and NMF can potentially recover the correct basis by interpolating the heatmaps, this may be a minor concern.

**m.4**  There are numerous issues with the notations and definitions. In future revisions, it would be beneficial to invest more effort in ensuring clarity for readers. Here are the identified errors and proposed corrections:
- The notation $ a(x| i, k) $ is incorrect as attribution method is not a conditional probability. You could either pass $(i,k)$ as arguments or attach them to the attribution function $a^{i}_k$.
- It's unclear if $i$ or $k$ is ignored for "layer-specific" attributions. Additionally, clarification is needed on what you call "layer-specific methods".
- $x_1, x_2 \in Y$ I propose you $(x_1, x_2) \in \mathcal{X}_c^2$ (page 5)
- Using $M$ for cluster when $M$ is already used could lead to confusion.
- Equation 4 lacks a closing parenthesis.
- Equation 5 also lacks a closing parenthesis.
- In Equation 6, $x$ should perhaps be bolded. Additionally, clarification is needed on what $P$ represents.
- When referring to "the matrix" produced by NMF, such as the "components" matrix, precise terminology is necessary. You could specify whether you are referring to the loading, coefficient, dictionary, codebook, or atoms matrix (or provide the correct shape). NMF produce 2 matrices (page 4).
- It would be beneficial to specify the method or formulas used to compute neuron attribution, as it appears to be an important parameter in the method that impacts the number of neurons.
- Instead of defining $SG$ if it's only used once, you could directly use $ a(x, i) \forall i \in \mathcal{I} $ for clarity (with $\mathcal{I}$ the indices of neurons directly, no need of layer.

**m.5**: Regarding the threshold, I strongly advise using percentiles instead of arbitrary values, as many values could be concentrated at the same spot (page 15).

**m.6**: Please consider increasing the font size of Figure 2, Figure 3, and Figure 5 for improved readability.

In conclusion, while the paper presents an interesting contribution to a real problem, addressing these points in a revised version for the next workshop or conference would significantly enhance its quality and clarity. Good job again.


[1] Sanity check for saliency maps, Adebayo & al.

[2] Towards faithfully interpretable NLP systems: How should we define and evaluate faithfulness?, Jacovi & al.

[3] What I Cannot Predict, I Do Not Understand: A Human-Centered Evaluation Framework for Explainability Methods, Colin & al.

[4] HIVE: Evaluating the Human Interpretability of Visual Explanations, Kim & al.

[5] RISE: Randomized Input Sampling for Explanation of Black box, Petsiuk & al.

[6] Evaluating and Aggregating Feature-based Model Explanations, Bhatt & al.

[7] Quantus: An Explainable AI Toolkit for Responsible Evaluation of Neural Network Explanations and Beyond, Hedström & al.

**Reason For Not Giving Higher Score:**

See **M.1** and **M.2**.

**Reason For Not Giving Lower Score:**

I believe the article is already in a good state, the method is already an interesting solution. However the paper would require a major rewrite and a better evaluation measure. I would love to see a revision of this paper and would encourage the authors to continue.

**Reviewer Domain:**

machine learning

---

### Official Review · Reviewer_Hvr5 · 2024-02-29
**Need more polishing on presentation but interesting work**

**Rating:** 2
**Fit:** 3
**Confidence:** 2

**Workshop Review:**

This paper proposes a new method for generating interpretable visual explanations for deep neural image classifiers. The proposed approach aims to reduce explanation complexity while identifying important concepts for bird classification. The authors also collected annotations from human experts for evaluating the alignment between the attribution maps produced from different explanation methods and feature maps corresponding to expert-defined concepts for bird classification. The method outperforms other baseline methods around the same level of explanation complexity on IoU scores with expert annotations. The work introduces a novel method and a new evaluation setup to measure the alignment between explanation methods and expert intuitions, and could thus invite more interest on this topic from the community.

**Reason For Not Giving Higher Score:**

The writing is technically solid but need more polishing on notations and figures:
1. There are typos in equation 1, 2, and 5 where the notations are missing brackets.
2. The fonts in the figures are too small to read.
3. The marks for different methods in fig 3 is hard to differentiate.
4. The choice of making CRP-288 a large square could be distracting for readers. It is ok to not make it this big since you already have a subplot (f) to show comparisons of explanation complexity.

As for technical details for the methods, it could benefit readers more if you can describe more about why you use NNMF and DBSCAN for  compressing concepts and clustering features. Do you have ant exploration over other techniques? Is your explanation recipe sensitive to these middle steps?

**Reason For Not Giving Lower Score:**

The experiment setup is complete and the writing is clear for both methods and evaluation. Fairly amount of baselines are implemented and evaluated in the paper. I also appreciate the effort for collecting annotations from expert to evaluate alignment. It would be beneficial to the community if such dataset could be published.

**Reviewer Domain:**

machine learning

---

### Decision · Program_Chairs · 2024-03-02

Accept (Poster)